# Load Measures in Training/Match Monitoring in Soccer: A Systematic Review

**DOI:** 10.3390/ijerph18052721

**Published:** 2021-03-08

**Authors:** Mauro Miguel, Rafael Oliveira, Nuno Loureiro, Javier García-Rubio, Sergio J. Ibáñez

**Affiliations:** 1Training Optimization and Sports Performance Research Group (GOERD), Sport Science Faculty, University of Extremadura, 10005 Caceres, Spain; sibanez@unex.es; 2Sport Sciences School of Rio Maior, 2040-413 Rio Maior, Portugal; rafaeloliveira@esdrm.ipsantarem.pt (R.O.); nunoloureiro@esdrm.ipsantarem.pt (N.L.); 3Life Quality Research Centre (CIEQV), Polytechnic Institute of Santarem, 2040-413 Rio Maior, Portugal; 4Research Centre in Sport Sciences, Health Sciences and Human Development, 5001-801 Vila Real, Portugal

**Keywords:** soccer, training, match, internal load, external load

## Abstract

In soccer, the assessment of the load imposed by training and a match is recognized as a fundamental task at any competitive level. The objective of this study is to carry out a systematic review on internal and external load monitoring during training and/or a match, identifying the measures used. In addition, we wish to make recommendations that make it possible to standardize the classification and use of the different measures. The systematic review was carried out according to the Preferred Reporting Items for Systematic Reviews and Meta-Analyses (PRISMA) guidelines. The search was conducted through the electronic database Web of Science, using the keywords “soccer” and “football”, each one with the terms “internal load”, “external load”, and “workload”. Of the 1223 studies initially identified, 82 were thoroughly analyzed and are part of this systematic review. Of these, 25 articles only report internal load data, 20 report only external load data, and 37 studies report both internal and external load measures. There is a huge number of load measures, which requires that soccer coaches select and focus their attention on the most useful and specific measures. Standardizing the classification of the different measures is vital in the organization of this task, as well as when it is intended to compare the results obtained in different investigations.

## 1. Introduction

In recent years, soccer coaches, members of technical staff and sports scientists have given particular attention to training and match monitoring. The amount of work performed by soccer players in training and match, as well as the consequent individual responses, positively or negatively affect their performance, leaving them more or less vulnerable to injury. Thus, the load monitoring process should assist coaches’ decision making about the players’ availability for training and competition [1], having as main objectives the improvement of performance and injury prevention [2,3,4]. For this reason, and also because of technological and analytical method developments [2], nowadays there is a huge set of load measures obtained through the use of telemetry and global positioning systems (GPS), among other microtechnologies [5].

Load measures can be categorized as either internal or external [1], depending on whether they refer to measurable aspects occurring internally or externally to the athlete [6]. External loads are objective measures of work performed by the athlete during training or competition [1], which are determined by the organization, quality, and quantity of exercise (training plan) [6]. Most common measures of external load include power output, speed, acceleration, time–motion analysis [1], and deceleration. By contrast, internal loads are defined as the relative biological (both physiological and psychological) stressors imposed on the athlete during training or competition [1], reflecting the psychophysiological responses that the body initiates to cope with the requirements elicited by external load [6]. Measures such as heart rate (HR), blood lactate (BLa), and rated perceived exertion (RPE) are commonly used to assess internal load [1].

It is unanimously believed that an integrated approach, rigorous and consistent, combining the use of internal and external loads, provides more significant information about the stress caused in soccer players than interpretations based on isolated data [1,3,6], and it is also recognized that this information should be simplified, with reporting limited to a few key metrics [1].

Impellizzeri [6] clarifies the importance of integrating both types of load, exemplifying that the uncoupling between internal and external load may be used to identify how athletes are coping with their training program. Specifically, athletes who exhibit a lower internal load to standardized external load completed under similar conditions would be assumed to reflect increased fitness. By contrast, when the internal load is increased in this situation, the athlete may be losing fitness or suffering from fatigue.

Methods that directly quantify a unit of measure (e.g., HR, distance, speed, time) or are able to count occurrences or repetitions are easily interpretable and can be used to plan and prescribe training, as well as to evaluate demands of competition. The use of composite or derivate methods, usually measured in arbitrary units (AU) (e.g., training impulses derived from HR), metabolic power (derived from locomotor acceleration and deceleration), player load (derived from accelerometer acceleration) and session rated perceived exertion (sRPE) (derived from perception of effort), adds more complexity to the interpretation of results but may bring more insight if analyzed correctly [1].

However, there is currently no consensus as to which variables are most useful or, indeed, how to analyze the longitudinal data of a diverse squad of players [2]. Reina [7] reinforces the importance of conducting systematic reviews about training/match monitoring with increasing attention given to this task, and therefore, there are a lot of data to collect and organize.

The identification of internal and external load measures used by investigations that use training or match as environment of monitoring can provide answers about which variables to include in an integrated approach. Thus, this systematic review aims to compile and order all the load measures used in soccer training/match monitoring, systematizing them.

## 2. Methods

The systematic review was carried out according to the Preferred Reporting Items for Systematic Reviews and Meta-Analyses (PRISMA) guidelines. In the present study, the search strategy followed by Sarmento [8] was adopted. The research was conducted on November 8, 2019, through the electronic database Web of Science (WOS). We chose WOS database because it is a research engine that groups other databases such as (1) Web of Science Core Collection; (2) Current Contents Connect; (3) Derwent Innovations Index; (4) KCI—Korean Journal Database; (5) Medline; (6) Russian Science Citation Index; and (7) SciELO Citation Index. The keywords “soccer” and “football” were used, associating each of them with the terms “internal load”, “external load” and “workload”. Therefore, we did six searches: “soccer” + “internal load”; “soccer” + “external load”; “soccer” + “workload”; “football” + “internal load”; “football” + “external load”; and “football” + “workload”.

### 2.1. Search Strategy: Inclusion Criteria and Process of Selection

Our analysis elected to review experimental and descriptive studies that met the following inclusion criteria: (1) published in peer-reviewed journals; (2) written in English; (3) report data about load monitoring in training and/or a match; (4) participants are male soccer players competing at the regional and/or national level. Studies involving (1) non-federated players, (2) female soccer players, (3) other sports, such as futsal, were excluded from the review, as well as studies that (4) did not present data collection of internal and/or external load, or that (5) reported exclusively data collected in specific exercises, such as small-sided games. If there was no agreement between the authors regarding the inclusion of any article, their inclusion/exclusion was discussed in order to reach a consensus.

It is necessary that there be reliability in the process of recording data in systematic reviews [9]. For this, consensus agreement between the coders was used. Two independent reviewers individually examined citations and abstracts to identify articles that potentially met the inclusion criteria. In these articles, a full-text analysis was carried out by the two reviewers to determine whether they met the inclusion criteria. Disagreements about the inclusion criteria were resolved through discussion between the authors, with all final decisions resulting from a joint analysis process.

### 2.2. Quality of the Studies and Data Extraction

To assess the quality of the studies, the risk-bias quality form used by Gómez-Carmona [5], Reina [7], Sarmento [8] and García-Santos [10], adapted from the original version developed by Law [11], was adopted. This evaluation is composed of 16 items and was performed by two researchers, with valuable expertise on this topic.

Articles were evaluated according to their objectives (item 1), relevance of background literature (item 2), adequacy of study design (item 3), sample studied (item 4 and 5), use of informed consent procedure (item 6), outcome measures (items 7 and 8), description method (item 9), significance of results (item 10), analysis (item 11), practical importance (item 12), description of dropouts (item 13), conclusions (item 14), practical implications (item 15), and limitations (item 16). The 16 quality criteria were rated on a binary scale (0/1), of which one of these criteria (item 13) presented the option: “If not applicable, assume 3”. The introduction of this option for item 13, “Were any dropouts reported?”, as justified by Sarmento [8], occurred because in some studies the investigators were not required to report dropouts (item 13).

The introduction of the option “not applicable” allowed a correct score of the article, eliminating the negative effect of assuming the value 0 on a binary scale, when in fact that item was not applicable to that study. As in the studies by Gómez-Carmona [5], Reina [7], Sarmento [8] and García-Santos [10], to make a fair comparison between studies with different designs, the percentage score was calculated as a final measure of methodological quality. For this, the sum of the score of all items was divided by the number of relevant scored items for that specific research design. All articles were qualified according to their score: low methodological quality, with 50% or less; good methodological quality, between 51% and 75%; and excellent methodological quality, greater than 75%. 

## 3. Results

### 3.1. Search Selection and Inclusion of Publications

The initial research identified 1220 titles in the electronic database Web of Science and another 3 titles in the electronic database ResearchGate. All records were exported to a bibliography management software (Endnote Web), with the duplicates being eliminated automatically (533 references). The 690 existing articles were subject to evaluation of the title, resulting in 413 exclusions from the database, and the remaining 277 articles were analyzed in the abstract. The analysis of the abstracts resulted in the elimination of 147 studies, so the full text of 130 articles was read, of which 48 were rejected due to the lack of relevance to the objectives of this study. The lack of relationship between studies and training/match monitoring proved to be the main reason for exclusions (*n* = 35). The integration of the female gender (*n* = 4), non-federated athletes (*n* = 6), and the other team sports (*n* = 3) in the studies was the reason for excluding the remaining articles. After this procedure, 82 articles were thoroughly analyzed and are part of this systematic review (Figure 1).

### 3.2. Quality of the Studies

Through the kappa index, as used in other reviews [5,7,8,10], a value of 0.755 was obtained for interobserver reliability, indicating a substantial agreement between observers [12]. The quality of the articles included in this review is confirmed by the following indicators: the average quality score of the articles was 85.7%; 75 articles had a score above 75%, and below 100% (excellent methodological quality); 7 articles exposed scores between 51% and 75% (good methodological quality); and no article had a score equal to or less than 50% (low methodological quality). The main reasons for the absence of maximum scores were due to the non-detailed description of the sample, the non-justification of the sample size, and/or the non-detailed description of the intervention.

### 3.3. Characterization of Studies

Table 1, Table 2 and Table 3 describe the main characteristics of the 82 articles including the analysis, which were published between 2004 and 2019, and the sample varied between 6 and 1200 subjects. From a total of 82 articles, 25 articles report only internal load data (Table 1), 20 report only external load data (Table 2), and 37 studies report both internal and external load measures. Of these studies, 78 evaluated athletes who competed at the national level, and in the other study, the level of the respective participants is not clarified. The monitoring period varies between 3 and 460 training sessions, and between 2 and 79 matches, with some studies only indicating the monitoring period (e.g., 30 weeks).

Considering age groups, 53 studies were carried out evaluating only adult players, 10 studies only U19, 4 studies only U17, 3 studies exclusively U15, and in 4 studies the age of soccer players is not presented. Additionally, 8 studies evaluated players from two or more age groups. In 27 studies, training was used exclusively as a monitoring condition, in 19 studies only the competition was used and in 36 studies both conditions were assessed.

Of the 27 studies that only evaluate training load, 9 studies examined only internal load measures, 2 studies only external load measures, and 16 studies both internal and external load. Of the 19 studies that assessed only the match load, 6 studies analyzed both internal and external load measures, 8 studies monitored only the external load, and 5 studies examined only the internal load. Of the 36 articles that evaluated both conditions (training and match load), 11 studies examined only the internal load, 10 studies measured only the external load, and 15 studies included the internal and external load.

The load measures are divided into 8 categories: “Heart Rate” (Internal Load), “Questionnaires and Inventories” (Internal Load), “Biomarkers” (Internal Load), “Distances” (External Load), “Training and Match Participation” (External Load), “Metabolic Power” (External Load), “Impacts” (External Load), and “Accelerations and Decelerations” (External Load). Figure 2 shows the number of studies that used each category of load measures, establishing a relationship regarding the utilization of the different categories of measures.

## 4. Internal Load Measures

The internal load measures were grouped according to their typology. Figure 2 shows the division of the different measures by each of the three categories.

### 4.1. Heart Rate

Heart rate (HR) is the number of heart beats per minute (bpm), and its monitoring has become a popular method for training control by measuring exercise intensity [95].

#### 4.1.1. Averages and Peaks

The average heart rate (HR_MEAN_) is determined in absolute (bpm) [26,37,60] and relative values (%HR_MAX_) [15,26,37,68,69,72,78,87,90,92,93]. Additionally, Campos-Vázquez [15] also measured the peak heart rate (HR_PEAK_) in relative values (%HR_MAX_) to assess the training sessions’ intensity.

#### 4.1.2. Intensity Zones

The intensity zones correspond to the division of the HR by intensity zones, measuring the activity time by zone. Several studies evaluate this measure, mostly in absolute values (min). However, there are differences regarding the division of the zones themselves. Wrigley [37] delimited HR assessment in six zones: <50%HR_MAX_, 51% to 60%HR_MAX_, 61% to 70%HR_MAX_, 71% to 80%HR_MAX_, 81% to 90%HR_MAX_, and >90%HR_MAX_, while Geurkink [74] evaluated the same zones with the exception of <50%HR_MAX_. Abade [58] and Coutinho [67] divided HR analysis into four zones: <75%HR_MAX_, 75% to 84.9%HR_MAX_, 85% to 89.9%HR_MAX_, and ≥90%HR_MAX_. Zurutuza [94] differentiated three zones of intensity: 50% to 80%HR_MAX_, 80% to 90%HR_MAX_, and >90%HR_MAX_. Campos-Vázquez [15] only quantified uptime above 80%HR_MAX_, Fullagar [72] exclusively measured uptime above 85%HR_MAX_, while Akenhead [59], Campos-Vázquez [16], and Stevens [91] only analyzed the activity time above 90%HR_MAX_. On the other hand, Silva [90] measured this measure in absolute (min) and relative values (%min), dividing the intensity in three zones: >70%HR_MAX_, >80%HR_MAX_, and >85%HR_MAX_.

#### 4.1.3. TRIMP Methods

Banister training impulse [96], Banister TRIMP, was established to quantify the internal load of a training session. This method considers the intensity (maximum heart rate, HR_MAX_; resting heart rate, HR_REST_; and average heart rate, HR_MEAN_) and exercise duration, *T*, using a coefficient, *y*, which relates heart rate and blood lactate during incremental exercise. The total load value, TRIMP, is expressed in arbitrary units (AU). This measure was used by Akubat [13], Impellizzeri [25], Scott [89], and Silva [90]. Since then, other authors developed methods for quantifying the total internal load that could provide more specific and individual responses:

Lucía’s TRIMP [97] justifies the evaluation of training load according to ventilatory thresholds (VT). This method, which divides the exercise intensity according to the heart rate reference values obtained in the cycle ergometer test, considers three zones: “light intensity” (<VT^1^), below 70%VO2_MAX_; “moderate intensity” (VT^1^–VT^2^), between 70 and 90%VO2_MAX_; and “high intensity” (>VT^2^), superior to 90%VO2_MAX_. Each zone is associated with a coefficient, 1, 2, and 3, respectively. The activity time, in minutes, in each zone is multiplied by the respective coefficient and added to obtain a total load value, expressed in AU. This measure was used by Impellizzeri [25].

Stagno TRIMP [98] directly evaluates the blood lactate profile instead of using a generic equation that reflects a hypothetical profile, obtaining a standard curve of response to increased exercise intensity. Five HR zones are then defined around the lactate threshold and onset of blood lactate accumulation (OBLA), 65–71%HR_MAX_, 72–78%, 79–85%, 86–92%, and 93–100%, with the respective weights 1.25, 1.71, 2.54, 3.61, and 5.16. The activity time, in minutes, in each HR zone is multiplied by the respective weighting to determine the total internal load, expressed in AU. This measure was used by Campos-Vázquez [15], Leiper [26], and Brink [62]. Recently, this calculation was modified by Akubat [13], who through the use of an exponential formula generated from the pooled data of all players, but without breaking up the subsequent equation into zones, called it Team TRIMP.

Individualized TRIMP, TRIMPi [99], which, contrary to the methods used by Banister [96] and Stagno [98], had as a weighting factor the physiological response of each athlete to exercise. To evaluate this factor, all athletes are subjected to a maximum test to determine the individual blood lactate concentration profile—blood lactate concentrations were plotted against running speeds and fractional HR elevation, and individual blood lactate concentration profiles were identified via exponential interpolation. Thus, *TRIMPi* = *T* × [(HR_MEAN_ − HR_REST_)/(HR_MAX_ − HR_REST_)] × *yi*, where *yi* reflects the profile of the standard curve of blood lactate response to increased exercise intensity. The *yi* values are calculated for each subject. The total load value, TRIMPi, is expressed in AU. This measure was used by Akubat [13] and Manzi [30].

Edward’s training load [100] includes a modification in the calculation of training impulses that simplifies the quantification of interval training. The activity time, in minutes, in each of the five HR zones is calculated and multiplied by a factor responding to each zone (50–60%HR_MAX_ = 1; 60–70% = 2; 70–80% = 3; 80–90% = 4; and 90–100% = 5). The results are then added together to determine a total internal load value, in AU. This measure was used by Campos-Vázquez [15], Impellizzeri [25], Leiper [26], Vahia [36], Casamichana [44], Condello [65], Fitzpatrick [71], Geurkink [74], Scott [89], Silva [90], and Zurutuza [94].

### 4.2. Biomarkers

The term “biomarker”, a portmanteau of “biological marker”, refers to a broad subcategory of medical signs that can be measured accurately and reproducibly [101]. A joint venture on chemical safety, the International Programme on Chemical Safety, led by the World Health Organization (WHO) and in coordination with the United Nations and the International Labour Organization, has defined a biomarker as “any substance, structure, or process that can be measured in the body or its products and influence or predict the incidence of outcome or disease” [102].

#### 4.2.1. Blood

The lactate produced during high-intensity exercises is simultaneously oxidized or transported from the production places to various tissues such as the heart, liver, kidneys, and muscle fibres for later oxidation [103], so this biomarker has been used to measure physiological stress imposed on soccer players. Blood lactate concentration (BLa) has been proposed as a measure of endurance fitness, but also as a means of standardizing training intensity. The steady-state exercise intensity that elicits a lactate concentration of approximately 4 mmol/L has been suggested as the most favourable to induce optimal physiological adaptations for resistance events [104]; however, the number of factors that affect the way lactate accumulates, independent of exercise intensity, make the importance of the lactate threshold less definitive, thus limiting its usefulness in monitoring and prescribing training intensity [105]. Aslan [60] collected a blood sample to measure the BLa in the first minute of the match, while Iacono [69] obtained the sample three minutes after the end of the training session and match.

Creatine kinase (CK), or creatine phosphokinase (CPK), is an important enzyme in the energy metabolism of skeletal muscle, which is usually present in the blood only in small concentrations. In soccer, this biological marker is used as a measure of muscle damage [57]. Wiig [57] collected blood samples 1 h before, and 1 h, 2 h, 48 h, and 72 h after the end of the match, having analyzed the CK concentration. Oliveira [83] measured the CK concentration in the plasma 48 h before competition.

Myoglobin, a heme-containing globular protein, is found in abundance in myocyte cells of the heart and skeletal muscle and is often referred to as an oxygen storage molecule or as an extra reserve of oxygen [106]. Practically null in terms of assessing the internal load in soccer, this variable was used in the study by [57] to measure muscle damage, with blood samples taken 1 h before, and 1 h, 24 h, 48 h, and 72 h after the end of the match.

#### 4.2.2. Saliva

Saliva sampling has rapidly developed as a tool for the assessment of biomarkers associated with physical performance [107]. Participating in high-intensity activities, with high demands and/or volume over a long period, can cause reductions in salivary immunoglobulin (SIgA) concentrations. SIgA can be used as an additional objective tool in training monitoring and quantifying workload [3], in order to avoid infections in the upper respiratory tract (URTI) [86]. In addition, the results obtained by Freitas [20,21] suggest that the evaluation of SIgA, in conjunction with the sRPE method, can be an insightful approach for coaches and their technical staff to assess the magnitude of training loads and the demands of the competition, contributing to adjust training plans. Figueiredo [70] assessed the SIgA concentration of this antibody 10 min before warming up and 10–15 min after the end of the match. Owen [86] measured the SIgA concentration 30 min before the start of the training session and just after its end.

### 4.3. Questionnaires and Inventories

The use of questionnaires to assess exercise and physical activity, particularly in large populations, is popular because its administration is easy and economical and does not affect training [105].

#### 4.3.1. Rating of Perceived Exertion

The perception of exertion is an important measure of an individual’s degree of physical strain [108], with Borg’s subjective rating of perceived exertion (RPE) being developed to allow the athlete, answering the question “How difficult/intense was the session?”, to subjectively assess their feeling regarding exercise, considering their own levels of physical fitness and fatigue [109]. Currently, as stated by Pescatello [110], there are two widely used RPE scales: the original Borg scale, which classifies the exercise intensity from 6 to 20 [26,60,62,86], and the modified scale, which measures from 0 to 10 [22,23,73,75,82,88]. In an attempt to simplify the training load quantification, Foster [111] introduced the use of the session rating of perceived exertion (sRPE) instead of using HR data or measuring the intensity of the session, or the type of exercise performed. The sRPE, obtained after the completion of training and/or the match, classifies the general difficulty of the session by multiplying the RPE by the duration of the exercise, in minutes [111] and, based on the scale of 0 to 10, has been widely used in the evaluation of internal load, both in training and in competition [7,9,10,12,13,14,15,16,17,18,19,20,24,26,28,29,30,31,32,39,61,65,66,67,68,69,70,72,74,75,77,78,79,80,81,84,85]. On the other hand, Coppalle [66] and Owen [86] used the 6–20 scale to determine sRPE. The application of this inventory has often occurred shortly after the end of the training session and/or match (15 to 30 min); however, it was applied by Owen [86] in the morning after training, in order to ensure that the perceived exertion reflected the whole session and not the last effort.

Lately, some studies have separately assessed the perceived cardiorespiratory (RPE_RES_) and muscular exertion (RPE_MUS_) [14,27,28,61,62,94], and a scale of 0 to 100 has also been used for this purpose [14], in which the technical demand (RPE_TECH_) is also assessed.

#### 4.3.2. Wellness and/or Recovery

One of the questionnaires that aims to assess the state of psychophysiological recovery of athletes is the Total Quality of Recovery (TQR) scale [112]. The use of a TQR scale makes it possible to monitor, and potentially accelerate, the recovery process simply by providing a more complete understanding of the actions necessary for achieving a total recovery [112]. Players perform TQR by answering the question “How recovered do you feel?” on one of two possible scales, 6 to 20 or 0 to 10. This questionnaire was applied by Campos-Vázquez [16] and Zurutuza [94] before the start of the training session and the match, and by Howle [24] 48 h after each match. Gjaka [22] modified this questionnaire to assess the level of recovery, having also submitted it to athletes before each training and match.

The Hooper Index is another questionnaire that subjectively assesses the feeling of well-being in relation to fatigue, stress level, muscle pain (DOMS), and quality of sleep [113]. Each of these parameters is measured separately before the training session or match, the index being the sum of the four indicators. These classifications use a scale of 1 to 7, from “very, very low/good” (point 1) to “very, very high/bad” (point 7) [114]. Clemente [18], Haddad [23], and Oliveira [84] applied this questionnaire before the beginning of each training session.

Recently, Howle [24] and Owen [86] customized this questionnaire to establish an image of individual daily well-being, modifying the scale used and some of the parameters evaluated. The questionnaire, with weightings from 1 (very poor) to 5 (excellent), includes questions about energy level, quality of sleep, readiness to train, and pain in lower body, allowing the sum of the partials to obtain an insight about the welfare state of the players before each training session. This questionnaire was applied before each training session, with Owen [86] trying to get answers about the previous training day.

Malone’s well-being questionnaire [29] is also an adaptation of the Hooper Index, assessing the feeling of well-being in relation to muscle pain, sleep quality, fatigue, stress, and energy level, and is applied before each training session. Athletes respond on a 7-point Likert scale, from 1 (strongly disagree) to 7 (strongly agree). The five individual well-being responses are added together to obtain an overall well-being score perceived by the athlete, with a maximum well-being score of 35 AU [29].

The Recovery–Stress Questionnaire for Athletes (RESTQ–Sport) was developed to measure the frequency of current stress symptoms, along with the frequency of activities associated with recovery. Through the simultaneous assessment of stress and recovery, it is possible to obtain a differentiated image of the current state of recovery-stress [115]. In this questionnaire, the interviewee indicates, on a Likert scale with values ranging from 0 (never) to 6 (always), the frequency with which he participated in activities or experienced relevant recovery/stress states. The questionnaire considers 19 items: general stress, emotional stress, social stress, conflicts/pressure, fatigue, lack of energy, somatic complaints, success, social relaxation, somatic relaxation, general well-being, sleep quality, disturbed beaks, burnout/emotional exhaustion, fitness/injury, fitness/being in shape, burnout/personal accomplish, self-efficacy, and self-regulation. It was applied by Fullagar [72] before each training session.

## 5. External Load Measures

The external load measures were grouped according to their typology. Figure 2 shows the division of the different measures by each of the five categories.

### 5.1. Distances

Locomotor activities, such as the total distance covered (TDC), high-speed running distance covered, or sprinting distance covered, are common external load metrics used by sport scientists [116]. The importance of studying locomotor activities was evidenced by McLaren [117] when he stated that the internal responses to training and match are strongly associated with the amount of running completed, rather than the myriad other external load measures typically monitored in team-sport athletes.

#### 5.1.1. Total Distance Covered

The total distance covered (TDC) is one of the most used external load measures in the evaluation of the amount of work developed by the players in training and competition, being measured in absolute (m) [29,32,38,39,42,43,44,45,46,47,48,49,50,51,52,53,55,57,58,59,60,64,66,68,70,71,72,73,74,76,77,78,80,83,84,85,86,87,88,89,90,91,93,94] and relative values (m/min [38,44,46,48,53,58,67,68,69,70,72,74,78,84,86,89,90], m/15 min [93], m/h [44], and %, represented as a % of the highest data reached in the match [54]).

#### 5.1.2. Distance Covered per Zone or Thresholds

The distance covered per speed zone is one of the preferred variables to assess the performance of soccer players. This measure, analyzed in absolute (m and min) [71,89] and/or relative values (m/min [38,53,69,80,90,92], %m [54,65,77,86,88], %m/min [54], and %min [44,50,65]), considers the division of the distance covered per speed zone, allowing a more detailed assessment of the work developed during training and/or the match. However, there is a great variability regarding the division and denomination of the zones. Aslan [60] delimited the distance covered in eight zones: “walking”, 0.0 to 6.0 km/h; “jogging”, 6.1 to 8.0 km/h; “low-intensity running”, 8.1 to 12.0 km/h; “moderate-intensity running”, 12.1 to 15.0 km/h; “high-intensity running”, 15.1 to 18.0 km/h; “low-intensity sprint”, 18.1 to 21.0 km/h; “moderate-intensity sprint”, 21.1 to 24.0; and “high-intensity sprint”, >24.0 km/h. Abade [58] and Coutinho [67] divided it into six zones: “zone 1”, 0.0 to 6.9 km/h; “zone 2”, 7.0 to 9.9 km/h; “zone 3”, 10.0 to 12.9 km/h; “zone 4”, 13.0 to 15.9 km/h; “zone 5”, 16.0 to 17.9 km/h; and “zone 6”, ≥18.0 km/h. Clemente [46,48] indicated four zones: “walking”, 0.0 to 6.9 km/h; “jogging”, 7.0 to 13.9 km/h; “running”, 14.0 to 20.0 km/h; and “sprint”, >20.0 km/h. Brito [43] also presented four zones: “low-intensity running”, <13.0 km/h; “high-intensity running”, 13.1 to 16.0 km/h; “very high intensity running”, 16.1 to 19.0 km/h; and “sprinting”, >19.1 km/h. Martín-García [52] exposed two zones: “high-speed running”, >19.8 km/h; and “sprinting”, >25.0 km/h. Giménez [75] demarcated six zones: “walking”, <2.2 m/s; “jogging”, 2.2 to 3.3 m/s; “low-speed running”, >3.3 to 4.2 m/s; “moderate-speed running”, >4.2 to 5.0 m/s; “high-speed running”, >5.0 to 6.9 m/s; and “sprint speed running”, >6.9 m/s. Jones [51] defined four zones: “low intensity”, <4.0 m/s; “moderate intensity”, 4.0 to 5.5 m/s; “high intensity”, 5.5 to 7.0 m/s; and “sprinting”, >7.0 m/s. Many other authors have presented different divisions and/or denominations [28,37,40,41,43,44,48,52,53,54,56,58,63,65,66,67,68,69,70,71,72,73,75,76,77,79,80,83,84,85,86,87,88,89,90,91,92]. In addition, the maximum distance [29,92] and average displacement [58,67,92] have been calculated at the “sprint” zone. The maximum speed reached by soccer players, in km/h [29,46,48,80] and in m/s [41,44,92], has also been evaluated.

Still in this category, the distance covered as a function of lactate thresholds is a measure used by Aslan [60], which consists of the individual assessment of running speed at lactate concentrations FBL_2_, FBL_2–4_, and FBL_4_, <2 mmol/L, 2 to 4 mmol/L, and >4 mmol/L, respectively, and consequent determination of the distance covered in each of the speed zones.

Finally, Bacon [39], Iacono [69], Fitzpatrick [71], Rago [88], and Zurutuza [94] used an individualized method to define one or more speed zones: Bacon [39] assessed the distance covered above 75% of maximum speed; Iacono [69] determined the sprint zone using the equation (25.2 (in-game *PV*) × 100, where *PV* stands for the peak speed achieved in match; Fitzpatrick [71] analyzed the distance covered above the maximal aerobic sprint (MAS) and ≥30% anaerobic sprint reserve (ASR), in meters and minutes (time spent at each zone); Rago [88] examined the distance covered ≥30% ASR, >80% MAS (“high-intensity activity”), 80.0% to 99.9% MAS (“moderate-speed running”) and 100% MAS to 29% ASR (“high-speed running”), in m and in %m; and Zurutuza [94] measured the distance covered above 80% of maximum speed, in m and in %m.

In addition, the absolute (*n*) [38,41,44,58,59,67,73,74,86] and relative number of efforts (*n*/min) [41,46,48] per speed zone were evaluated, as well as the time spent in each sprint/effort [92] and between sprints/efforts [44,58,67,76,92], both in absolute (s) and relative (%) values.

#### 5.1.3. Distance Ratios

Work-to-rest ratios are used to describe soccer players’ activity profiles [44,49,58,63,75,92]. To calculate this ratio, a speed zone is defined as “rest/recovery”, and another as “work/activity”, through which the distances covered in these zones are used to determine the ratio (division of the amount of work by the amount of rest). Casamichana [44,63] used a speed zone from 0.0 to 3.9 km/h as “rest/recovery” and a speed zone >4.0 km/h as “work/activity”. Abade [58] constituted three levels of ratio: 0.0 to 6.9 as “rest” and 7.0 to 9.9 km/h as “work”; 0.0 to 6.9 as “rest” and 10.0 to 15.9 km/h as “work”; and 0.0 to 6.9 km/h as “rest” and a speed zone >16.0 km/h as “work”. Giménez [75] indicated a speed zone from 0.0 to 2.0 m/s as “rest” and another >2.0 m/s as “work”. Suarez-Arrones [92] defined a zone from 0.0 to 7.0 km/h as “rest” and another >7.0 km/h as “work”.

Another ratio model is described by Clemente [49], who used the amount of distance covered in the microcycle and divided it by the load of the competition: “total distance ratio”; “running distance ratio”, 14.0 to 19.9 km/h; “high-speed running distance ratio”, 20.0 to 24.9 km/h”; and “sprinting distance ratio”, >25.0 km/h.

### 5.2. Accelerations and Decelerations

Acceleration is based on the change in GPS speed data, and it is defined as a change in speed for a minimum of 0.5 s, with a maximum acceleration of at least 0.5 m/s. The acceleration is considered complete when the player stops accelerating. The classification of speed zones is based on the maximum acceleration achieved in the acceleration period. The same approach is used in deceleration [80].

#### 5.2.1. Total Distance Covered Accelerating and Decelerating

Total distance covered during acceleration and deceleration was collected by Rago [55] and Akenhead [59] to characterize physical demands imposed by competition in soccer players against opponents of higher and lower qualitative levels.

#### 5.2.2. Distance Covered per Zone

The measurement of the distance covered at each acceleration and deceleration zone allows to measure the intensity of the displacements, regarding the starting and braking actions. This metric is analyzed in absolute (m) and relative (m/min, m/effort, and %) values. However, there is variability in the definition of the acceleration and deceleration zones, as well as in the classification of those same zones. Barron [40] divided the analysis into four deceleration zones and four acceleration zones: “zone 1” (deceleration), −5.0 to −20.00 m/s^2^; “zone 2” (deceleration), −4.0 to −5.0 m/s^2^; “zone 3” (deceleration), −2.0 to −4.0 m/s^2^; “zone 4”(deceleration), 0.0 to −2.0 m/s^2^; “zone 5” (acceleration), 0.0 to 2.0 m/s^2^; “zone 6” (acceleration), 2.0 to 4.0 m/s^2^; “zone 7” (acceleration), 4.0 to 5.0 m/s^2^; and “zone 8” (acceleration), 5.0 to 20.0 m/s^2^. Castagna [64] delimited two zones of deceleration, “high-intensity decelerations” (≤2.0 m/s^2^) and “very high-intensity decelerations” (≤−3.0 m/s^2^), and two zones of acceleration, “high-intensity accelerations” (≥2.0 m/s^2^) and “very high-intensity accelerations” (≥3.0 m/s^2^). Akenhead [59] defined three zones of deceleration and three zones of acceleration: “low decelerating”, −1.0 to −2.0 m/s^2^; “moderate decelerating”, −2.0 to −3.0 m/s^2^; “high decelerating”, <−3.0 m/s^2^; “low accelerating”, 1.0 to 2.0 m/s^2^; “moderate accelerating”, 2.0 to 3.0 m/s^2^; and “high accelerating”, >3.0 m/s^2^. Other authors presented different divisions and/or denominations [49,54,71,94]. Chrismas [45] added to this variable (distance in acceleration and deceleration >2.0 m/s^2^) the distance covered in high intensity (>5.5 m/s), in order to quantify the high-metabolic load (HML) that players experienced during training. The same method, in absolute (m) and relative (m/min) values, was used by Silva [90]; however, these authors considered the distance covered >14.4 km/h.

#### 5.2.3. Frequency of Efforts

The quantification of the number of accelerations and decelerations, total [70,75] or partial (by speed threshold), in absolute (*n*) and relative (*n*/min and %) [38,73,80,90], are measures used in recent studies.

In the division and denomination of these zones, there also is diversity. Curtis [68] indicated three zones of acceleration (“low intensity”, 0.0 to 1.99 m/s^2^; “moderate intensity”, 2.0 to 3.99 m/s^2^; and “high intensity”, >4.0 m/s^2^) and three of deceleration (“low intensity”, 0.0 to −1.99 m/s^2^; “moderate intensity”, −2.0 to −3.99 m/s^2^; and “high intensity”, <−4.0 m/s^2^). Stevens [91] divided the analysis of the number of accelerations and decelerations into four zones, two of acceleration (“medium efforts”, 1.5 to 3.0 m/s^2^ and “high efforts”, >3.0 m/s^2^) and two of deceleration (“medium efforts”, −1.5 to −3.0 m/s^2^ and “high efforts”, <−3.0 m/s^2^). Other authors have presented different divisions and/or classifications regarding these variables [38,52,73,74,75,76,77,85,87,90].

In addition to accelerations and decelerations above 2.0 m/s^2^, Iacono [69] added the number of sprints in the calculation of high-intensity efforts per minute (HIE/min). By contrast, Owen [53] sums the number of accelerations and decelerations above 4.0 m/s^2^ to define the amount of high-intensity efforts (HIE). Wiig [57] adopted the same strategy and summed the number of accelerations and decelerations above 2.5 m/s^2^ to obtain the number of HIE. Casamichana [44] and Jaspers [77] assessed the number, average, average duration, and maximum duration of repeated HIE (at least 3 efforts at a speed >13.0 km/h [44] or at least 3 sprints, high-magnitude accelerations (>3.5 m/s^2^), or a combination of both [77]—and with <21 s recovery between them).

#### 5.2.4. Accelerations and Decelerations Ratios

The acceleration and deceleration ratios were recently used by Clemente [49], who used the number of accelerations and decelerations (>3.0 m/s^2^) in the microcycle and divided it by the load of the competition itself.

#### 5.2.5. Player Load

The player load (PL) is based on the acceleration data that are recorded by triaxial accelerometers [118] and is one of the most used metrics to describe the external load [29,40,44,56,57,59,63,68,75,77,85,89]. This variable is considered a vector of magnitude that represents the sum of the accelerations recorded in the anteroposterior, mediolateral, and vertical planes [119]. Lately, other metrics that derive from PL have been used. The PL Slow quantifies the accelerations performed at a speed below 2.0 m/s [29]; the PL 2D omits the vertical accelerometer from the calculation, allowing a more precise quantification in relation to actions over short distances [94]; and the PL 1D consists in assessment of each axis of the movement in isolation [51,77]. These are measured in absolute (AU and g) [48,49] and relative (UA/min [62,77], g/min [46], and UA/m [51,77]) values.

#### 5.2.6. Exertion Index

The exertion index (EI) derives from the speed of movements on the playing field and it is calculated using three equations. These equations are the sum of the weighted instantaneous speed, the weighted cumulative speed over 10 s, and the weighted cumulated speed over 60 s [44]. This variable is evaluated in absolute, EI [75], and relative, EI/min [44], values.

### 5.3. Impacts

Impacts are often identified as values of maximum magnitude of the accelerometer, over 2 *g* over a period of 0.1 s, and they are reported as maximum and cumulative values over a specific period [120].

#### 5.3.1. Impacts Performed

The total (*n*) [38,58,80] and relative (*n*/min) [38,58,67,73,80,90] number of impacts is a variable used to measure the number of intense actions performed by soccer players.

#### 5.3.2. Impacts Performed per g Zone

In addition to the number of impacts, another measure is the distribution of the number of impacts by force zone, gArruda [38], Abade [58], and Coutinho [67] divided the impacts into six zones: “zone 1”, 5.0 to 6.0 g; “zone 2”, 6.1 to 6.5 g; “zone 3”, 6.6 to 7.0 g; “zone 4”, 7.1 to 8.0 g; “zone 5”, 8.1 to 10.0 g; and “zone 6”, ≥10.1 g. Gaudino [73] and Silva [90] analyzed only the number of impacts performed above 2.0 g. This division allows a more detailed analysis of the intensity of the body impacts.

#### 5.3.3. Dynamic Stress Load

Dynamic stress load is the total of the weighted impacts, which is based on accelerometer values of magnitude above 2.0 *g*. It weights the impacts using an approach similar to that used in the speed intensity or heart rate exertion calculations, with the key concept being that an impact of 4.0 *g* is more than twice as hard on the body as an impact of 2.0 *g* [120]. Weighted impacts are aggregated and organized at scale, expressed in AU to provide more useful values. This variable is quantified in absolute (AU) [70,73,80,90] and relative (AU/min) [73,80,90] values.

#### 5.3.4. Total Load

The total load gives the total of the forces on the player over the entire activity period based on accelerometer data alone. It uses the magnitude of the accelerometer values taken in three directions, sampled 100 times per second [120]. This metric was used in the study of Figueiredo [70].

### 5.4. Metabolic Power

Metabolic power (MP) has been proposed to provide an instant image of specific soccer activities [121]. This method considers acceleration and speed to define the profile of individual distances, and the time spent by players in power limits arbitrarily estimated and chosen [121,122]. This approach assumes that the energy produced by a player during a match is a direct result of the cost of the acceleration and the corresponding instantaneous speed [121]. Despite the name that implies metabolism of the athlete, it is mathematically derived from the speed–time profile and, therefore, remains as an external load measure [6].

#### 5.4.1. Averages

The average MP (W/kg) is known as the energy spent by the players per second, per kilogram of body weight, and it has been evaluated by different authors [52,56,64,73] to obtain information about the metabolic requirement experienced by soccer players during training and/or a match.

#### 5.4.2. Distance Covered per Zone

The assessment of the distance covered by the zone of MP exposes the amount of metabolic wear imposed by physical activity on the players. It was used by Martín-García [52], Gaudino [73], and Malone [80], who measured, in absolute (m) and/or relative (m/min) values, the distance covered in high metabolic power, >25.5 W/kg, as well as by Castagna [64] and Stevens [91], who determined the distance covered in “high power” and “high intensity”, respectively, ≥20.0 W/kg. Additionally, the activity time (in %) at different power zones was also assessed by Iacono [69], who divided the analysis into five zones: “low power”, 0.0 to 10.0 W/kg; “intermediate power”, 10.0 to 20.0 W/kg; “high power”, 20.0 to 35.0 W/kg; “elevated power”, 35.0 to 55.0 W/kg”; and “maximum power”, >55.0 W/kg.

In this category, the equivalent distance (ED) used in the study by Chrismas [45] represents the distance that the athlete would have covered at a constant pace using the total energy spent during the activity (training or match). *ED = W/ECcKT*, where *ED* is expressed in meters, *W* is the total energy expended (J/kg), *ECc* is the energy cost of running at a constant speed, which is assumed as 3.6 J/kg, and *KT* is a factor associated with the type of floor where soccer is played (=1.29) [121].

### 5.5. Training and Match Participation

In soccer, the density of the competitive period requires a careful periodization by coaches and their staff. The number of training sessions per microcycle, the duration of each training session, and the time of participation in the competition are variables that affect, positively or negatively, the physical fitness of the soccer players.

#### Frequency and Duration

The total exposure time to the match [57,83,85,87] or to the training sessions [29,47,70,74,77,78,82,83,84,85,91], in minutes, as well as the number of training sessions and matches [82] is quantified to assess the external load imposed on athletes.

## 6. Discussion

The purpose of the present study is to carry out a systematic review on internal and external load monitoring, in training/match, identifying the measures used. Simultaneously, we intend to order all the load measures used in soccer monitoring and systematizing them (describe them, grouping by categories, and standardize their structure/classification). In recent years, training and match load monitoring has received special attention from sports scientists, of which 55 articles included in this systematic review were published between 2017 and 2019. Through the analysis of all articles used in this systematic review, we verified, as stated in the Vanrenterghem [4] study, the existence of a colossal and increasing number of load measures at the disposal of soccer coaches. On the one hand, it increases the range of monitoring options, and on the other hand, it raises doubts about which measures are most valid, useful, and important to analyze. The technological evolution in the monitoring instruments not only allows improvements in the accuracy of the collected data, but also promotes the development of new measures and/or versions of existing measures. Moreover, as described by Akenhead [2], the multiplicity caused by the evaluation on the same load measure in terms of volume (absolute values) and intensity (relative values) of work is another reason that gives rise to the abundance of load measures. Foster [123] adds that the future of training monitoring might well be dominated by emerging technologies that allow new possibilities relative to the analysis of the external training load and, in that sense, there are very relevant works [1,2,3,4,6,123] in the theorization and conceptualization of monitoring purposes. However, there are few (practically non-existent) works that expose the load measures used in the training and match evaluation and that describe them systematically, providing those who start, or already work in this area of activity, a repository of basic knowledge.

The load categories most used to monitor training and match are “Distances” (53 studies), “Questionnaires and Inventories” (48 studies), “Accelerations and Decelerations” (35 studies), and “Heart Rate” (28 studies). Less used are the “Training and Match Participation” (13 studies), “Biomarkers” and “Metabolic Power” (both with 8 studies), and “Impacts” (7 studies). Regarding the load measures, the most used are “Distance Covered per Zone/Threshold” (50 studies), “Total Distance Covered (47 Studies), “Rating of Perceived Exertion” (45 studies), “Frequency of Efforts” and Player Load (both with 18 studies), “TRIMP Methods” (15 studies), “Frequency and Duration” (13 studies), “Average and Peaks” (12 studies), and “Intensity Zones” (11 studies). In these measures, the collected data are analyzed in absolute (e.g., m, AU, and min) and/or relative (e.g., %, m/min, and AU/min) values.

In agreement with Bourdon’s opinion [1] regarding the need to simplify the information for some main load measures, also corroborated by Akenhead [2], who verified that coaches, excluding match/training duration, record 7 ± 2 measurements in the monitoring tasks, we propose that the following measures should be considered: average heart rate (%HR_MAX_) [14,25,36,67,68,71,77,86,89,91,92]; heart rate intensity zones (min and %min) [15,16,37,58,59,67,72,74,91,94]; sRPE (AU and AU/min) [7,9,10,12,13,14,15,16,17,18,19,20,24,26,28,29,30,31,32,39,61,65,66,67,68,69,70,72,74,75,77,78,79,80,81,84,85]; Edward’s training load (AU) [15,25,26,36,44,65,71,74,89,90,94]; Hooper index (AU) [18,23,24,29,84,86]; player load 3D (AU, AU/min, and %AU) [29,40,44,56,57,59,63,68,75,77,85,89]; total distance covered (m, m/min, and %m) [29,32,38,39,42,43,44,45,46,47,48,49,50,51,52,53,55,57,58,59,60,64,66,68,70,71,72,73,74,76,77,78,80,83,84,85,86,87,88,89,90,91,93,94]; distance covered by speed zone (m, m/min and %m) [29,38,41,42,43,44,45,46,48,49,51,52,53,54,55,57,58,59,60,64,66,67,68,69,70,71,72,73,74,75,76,77,78,80,81,84,85,86,87,88,89,90,91,92,93]; distance ratios [44,49,58,63,75,92]; accelerations and decelerations(*n*, *n*/min and %*n*) [38,68,73,74,75,76,77,80,85,87,90,91]; and training/match duration (min) [29,47,57,70,74,77,78,82,83,84,85,87,91]. We identified five internal load and six external load measures among the most used in scientific articles included in this review, which include, as suggested by Bourdon [1], variables that directly quantify units of measurement, as well as composite methods, capable of globally evaluating the quantity and quality of training sessions and matches. When organizing a monitoring approach, we recommend that the inclusion of some of these measures should be considered. The importance attributed to the selected measures may vary from session to session, due to the alternation of physical regimes during the microcycle, or between the player positions that have different physical actions and demands during matches [14,40,43,54]. Altogether, the assessment of internal and external load measures will help coaches, sport scientists, and researchers to compare loads from different studies and to replicate different studies methodologies for their own exercise training sessions, planning, and periodization.

The internal load measures derived from biomarkers are not present among the most used mainly due to the constraints involved in their daily and systematic collection, both in training and in competition. This type of measures is useful and valid [60,69,105]; however, it is more appropriate for carrying out evaluations with less continuous nature.

We found that in some studies the term “training load” [16,17,20,22,28,33,65,83] is used as a parameter for assessing both the load suffered by athletes in training as in the match. However, we consider that this designation is not appropriate because it does not distinguish the medium of load monitoring, the training, or the match. Until a few years, training monitorization was only used in the context of training and the term “training load” was tolerable, nowadays, load monitorization is also used in competition and therefore, this designation becomes reductive. As applied by other studies [13,33,34,49,52,61,71,76,82,87], we recommend the use of the terms “training load” and “match load” (regardless of whether they are friendly or official) to differentiate where the load assessment is developed. When the assessed load corresponds to the sum of training and match load, we suggest applying the “workload” term [19,31,33,42,44,50,58,67,75].

All studies evaluating accelerations and decelerations use “m/s^2^” as the unit of measurement; however, the same does not happen with respect to the distance covered by speed zones. Most studies use “km/h” [29,49,64,66,67,68,71,73,74,77,83,86,87,89] as the unit of measurement and a smaller number of studies use “m/s” [45,51,59,75]. We suggest the use of “m/s” as a unit of measurement in detriment of “km/h”, and it is not just about converting “km/h” to “m/s”. Notice that Owen [86] considers “jogging” the distance covered at a speed between 7.3 and 14.3 km/h (if we convert to m/s, ≈2.0 to 4.0 m/s); Clemente [46] considers “jogging” the distance covered between 7.0 and 13.9 km/h (if we convert to m/s, ≈1.9 to 3.9 m/s); Giménez [75] considers “jogging” the distance covered between 2.2 and 3.3 m/s (if we convert to km/h, ≈7.9 to 11.9 km/h). In these three studies, performed with professional soccer teams, there is no consensus regarding the “jogging” zone. What are the reasons for this? There are several, but one of them we consider to be the use of different units of measurement (“m/s” and “km/h”). We recognize that the standardization of the unit of measurement used will allow us to improve the existing consensus regarding speed zones, just as the “m/s” unit also fits much better to what is used by soccer coaches when preparing their plans and when they intervene. For example, when a coach planning an exercise to improve “sprint” speed, where the exercise space is 20 m long, does it become simpler to control if he sets a goal to reach in 7.0 m/s or in 25.2 km/h? The issue is not the simple conversion from “km/h” to “m/s”; it is the utility and functionality of the unit of measurement. In summary, we suggest using the “m/s” unit when evaluating the distance covered by speed zone, and the “m/s^2^” unit when evaluating the acceleration/deceleration zones.

Furthermore, there are inconsistencies in the denomination and categorization of some measures, which may indicate uncertainties about the validity and usefulness of what is being examined, and they make it difficult to compare results between different investigations. Consequently, we consider its uniformity and systematization to be critical. Regarding heart rate, we observe that are differences with respect to the definition of heart rate intensity zones. Wrigley [37] found lower values of average HR (%HR_MAX_) for U18 compared to U16 and U14: in training, U18—69 ± 2%, U16—74 ± 1%, and U14—74 ± 2%; and in match, U18—81 ± 3%, U16—84 ± 2%, and U14—83 ± 2%. In the monitorization carried out by Iacono [69] in U19, mean HR values (%HR_MAX_), 84.5 ± 2.8%, were found in UEFA Youth League matches. Malone [78] assessed the average HR (%HR_MAX_) of the training sessions in the preparatory period and identified values of 70 ± 7%, with Silva [90] finding similar values in the same period of the season, 71.2 ± 5%. Torreño [93] measured, in an elite senior team, the average HR (%HR_MAX_) during the matches and obtained values of 86 ± 4.9%. Thus, considering the analysis of %HR_MAX_ and the permanent existence of variables that affect the intensity of training/match, we recommend the definition of four zones of intensity: <60.0%HR_MAX_ (Wrigley [37] and Geurkink [74] define 60% as the upper limit of one of their intensity zones), 60.0% to 74.9%HR_MAX_ (Wrigley [37] and Geurkink [74] define an intensity zone above 60%, while Abade [58] and Coutinho [67] define one below 75%), 75.0% to 89.9%HR_MAX_ (sum of two intensity zones presented by Abade [58] and Coutinho [67]), and ≥90%HR_MAX_ [15,37,58,59,67,74,91,94]. Additionally, in Abade [58] and Coutinho [67] studies, the intensity zones are called “zone 1, 2, 3, and 4”; however, this type of denomination does not clarify the intensity involved. We suggest, for the indicated intensity zones, to name them as “low intensity”, “moderate intensity”, “high intensity”, and “maximum intensity”, respectively. When we propose the use of these four zones, our intention is to standardize the evaluation parameters. How can we compare two studies performed in similar conditions [37,58] and the results obtained by them if, for example, one uses an intensity zone of 71.0% to 80.0%HR_MAX_ [37] and the other of 75.0% to 84.9%HR_MAX_ [58]? If the heart rate intensity zones are not consensual between studies, we will lose detail in the analysis and, consequently, the practical applications of the studies become dubious as there are different configurations regarding the definition of the intensity zones.

Concomitantly, when analyzing the speed zones, we verified that the distance covered above 4.0 m/s is called both “high-speed running” [80] and “high-intensity activity” [88]. In professional soccer, the distance covered above 5.5 m/s is defined as “high-speed running” [45]. Previously, we suggest the use of “intensity” for variables related to heart rate (e.g., intensity zones) or accelerations and decelerations (distance covered or number of efforts performed), and the use of “speed” when related to distance covered by different speed thresholds. Then, we propose the definition of six speed thresholds: “walking distance”, 0.0 to 2.0 m/s; “jogging distance”, 2.0 to 3.0 m/s; “running speed distance”, 3.0 to 4.0 m/s; “high-speed running distance”, 4.0 to 5.5 m/s; “very high-speed running distance”, 5.5 to 7.0 m/s; and “sprint distance”, a speed greater than 7.0 m/s. Subsequently, we recommend, as defined by Giménez [75], that a speed zone <2.0 m/s should be considered as a “rest” in the assessment of ratios related to the distance covered. If an elite European soccer player covers 107 ± 12 m/min [124], approximately 1.78 ± 0.2 m/s, the definition of a speed zone >2.0 m/s as “rest” includes a speed usually higher than the average running speed presented in competition by national teams and, therefore, can hide the part of the “work” developed. Furthermore, with regard to accelerations and decelerations, we recommend the definition of three zones used by Curtis [68]: “low intensity”, 0.0 to 2.0 m/s^2^; “moderate intensity”, 2.0 to 4.0 m/s^2^; and “high intensity”, greater than 4.0 m/s^2^.

## 7. Limitations

Some limitations were addressed when considering this research on the training/match load monitoring in soccer. Only 3 of 82 studies are based on the regional level. In this sense, the conclusions obtained mainly portray the load measures in elite soccer (in different age groups) and cannot be generalized to any competitive level. Finally, 65% of the articles included in this review present a sample exclusively composed of adult soccer players, which influences the choice and definition of the load measures evaluated in the training and/or match, and we do not differentiate the analysis by age group.

## 8. Conclusions and Practical Applications

In soccer, training and match load monitoring is recognized as a relevant task at any competitive level. Through this monitorization, the coaches and other members of the technical staff can base part of their decision making about the periodization, design, and application of the different types of planning (training exercise, training session, microcycle and mesocycle), and the individual and collective management of the team in training process and in the competition.

However, due to the inconsistencies examined in the criteria for identifying and systematizing various measures, it is critical to standardize their structure and classification. This will allow to have confidence about the validity and usefulness of what is being analyzed, as well as to promote the possibility of comparing the results of different investigations and, consequently, to increase and improve knowledge about this very sensitive subject.

This systematic review reveals the measures used in scientific articles that focus on internal and/or external load monitoring in training sessions and/or matches and it could be used as an instrument for the reorganization and standardization of various load measures. From the findings of the present systematic review, relevant practical applications should be considered:

(a)Nomenclature and Organization—“Training load” only represents the load assessed in the training sessions. “Match load” represents the evaluation of the load imposed by the games, with an official or friendly nature. “Workload” corresponds to the sum of training and match load. Additionally, to clarify the structure and classification of this activity, it is essential to use a standard nomenclature and order. The use of different names, or values, for the same variable causes entropy. We specifically indicate the nomenclature to be used, as well as the range of values that define each speed, acceleration/deceleration, and heart rate intensity zone;(b)Identification of Load Measures—Our study systematically describes all the load measures used by the articles included in this review, providing those who start, or already work in this area of activity, a repository of basic knowledge;(c)Selection of Load Measures—Due to the existence of an extraordinary number of load measures, it is essential that soccer coaches and/or sport scientists select and focus their attention on the most useful and specific measures. Based on the measures most used by the articles included in this review, we suggest a set of internal and external load measures to be considered in that selection;(d)Units of Measure—The use of the “m/s” unit when evaluating the distance covered by the speed zone, to the detriment of “km/h”, will improve the existing consensus regarding speed zones, as well as take on a more functional character;(e)Intensity vs. Speed—The use of “intensity” to variables related to heart rate or accelerations and decelerations, and the use of “speed” when related to distance covered by different speed thresholds.

## Figures and Tables

**Figure 1 ijerph-18-02721-f001:**
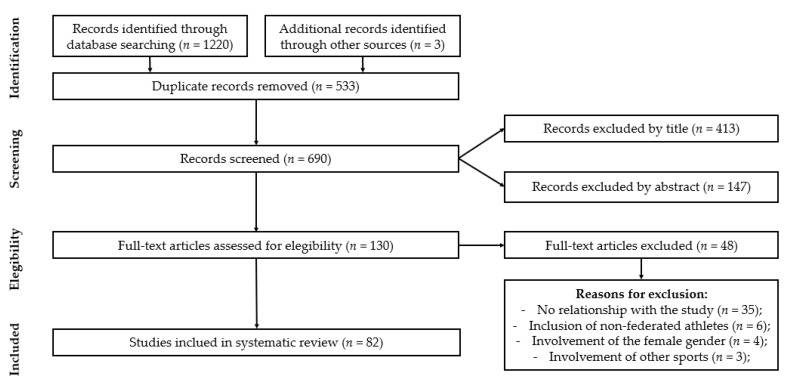
Article selection process flowchart.

**Figure 2 ijerph-18-02721-f002:**
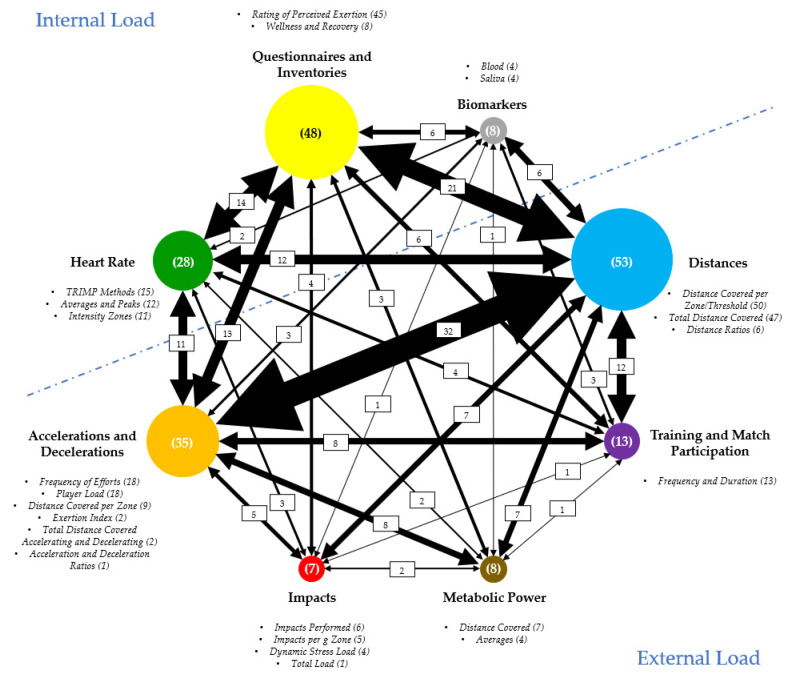
Load measures groups: quantities and relationships.

**Table 1 ijerph-18-02721-t001:** Characteristics of studies that evaluated only internal load measures.

Study	Level	Sample	Age	Condition	Duration	Quality
Akubat [13]	National	14	17.0 ± 1.0 year	Training Competition	24 Sessions 6 Matches	Excellent
Barrett [14]	National	32	25.0 ± 8.0 year	Competition	38 Matches	Excellent
Campos-Vasquez [15]	National	9	26.7 ± 4.5 year	Training	288 Sessions	Good
Campos-Vasquez [16]	National	12	27.7 ± 4.3 year	Training Competition	21 Sessions 7 Matches	Excellent
Cetolin [17]	National	18 12	U15—14.7 ± 0.5 year U19—18.9 ± 0.9 year	Training Competition	40 Sessions 3 Matches 45 Sessions 6 Matches	Excellent
Clemente [18]	National	35	25.7 ± 5.0 year	Training	192 Sessions	Excellent
Delecroix [19]	National	130	N/D ^1^	Training Competition	1 Season	Excellent
Freitas [20]	National	11	16.5 ± 0.5 year	Training Competition	4 Weeks	Excellent
Freitas [21]	National	26	15.6 ± 1.1 year	Competition	4 Matches	Excellent
Gjaka [22]	National	22	14.5 ± 0.3 year	Training Competition	12 Sessions 6 Matches	Excellent
Haddad [23]	National	17	18.2 ± 0.5 year	Training	21 Sessions	Excellent
Howle [24]	National	42	26.4 ± 5.1 year	Competition	37 Matches	Excellent
Impellizzeri [25]	N/D ^1^	19	17.6 ± 0.7 year	Training	27 Sessions	Excellent
Leiper [26]	National	79	18.0 ± 1.0 year	Training	38 Sessions	Excellent
Los Arcos [27]	National	40	N/D ^1^	Competition	2 Seasons	Good
Los Arcos [28]	National	24	20.3 ± 2.0 year	Training Competition	30 Weeks	Good
Malone [29]	National	48	25.3 ± 3.1 year	Training	460 Sessions	Excellent
Manzi [30]	National	18	28.4 ± 3.2 year	Training	8 Weeks	Excellent
McCall [31]	National	171	25.1 ± 4.9 year	Training Competition	1 Season	Excellent
Pinto [32]	National	20	16.8 ± 0.6 year	Competition	2 Matches	Excellent
Raya-González [33]	National	22	18.6 ± 0.6 year	Training Competition	141 Sessions 38 Matches	Excellent
Rowell [34]	National	23	23.3 ± 4.1 year	Training Competition	1 Season 34 Matches	Excellent
Saidi [35]	National	18	20.1 ± 0.4 year	Training	26 Sessions	Excellent
Vahia [36]	National	15	16.7 ± 1.0 year	Training	160 Sessions	Excellent
Wrigley [37]	National	8 8 8	U14—13.0 ± 1.0 yearU16—15.0 ± 1.0 yearU18—17.0 ± 1.0 year	Training Competition	6–8 Sessions 2 Matches	Excellent

^1^ Not defined (N/D).

**Table 2 ijerph-18-02721-t002:** Characteristics of studies that evaluated only external load measures.

Study	Level	Sample	Age	Condition	Duration	Quality
Arruda [38]	National	10	15.1 ± 0.2 year	Competition	5 Matches	Excellent
Bacon [39]	National	18 23	18.8 ± 1.2 year 17.0 ± 1.1 year	Training Competition	40 Weeks	Excellent
Barron [40]	Regional	38	17.3 ± 0.9 year	Competition	8 Matches	Excellent
Bendala [41]	National	25	26.5 ± 4.1 year	Training Competition	41 Weeks 9 Matches	Excellent
Bowen [42]	National	32	17.3 ± 0.9 year	Training Competition	2 Seasons	Excellent
Brito [43]	Regional	66	13.4 ± 0.5 year	Competition	9 Matches	Excellent
Casamichana [44]	National	27	22.8 ± 4.5 year	Training Competition	9 Sessions 7 Matches	Excellent
Chrismas [45]	National	6	26.0 ± 2.0 year	Training	247 Sessions	Excellent
Clemente [46]	National	14 15	19.2 ± 1.0 year 25.1 ± 3.9 year	Training	7 Weeks	Excellent
Clemente [47]	National	23	24.7 ± 2.8 year	Training Competition	47 Sessions 12 Matches	Excellent
Clemente [48]	National	18 24 23 24	25.4 ± 4.8 year 21.5 ± 2.5 year 23.0 ± 3.7 year 24.7 ± 2.9 year	Training Competition	5 Weeks	Excellent
Clemente [49]	National	27	24.9 ± 3.5 year	Training Competition	22 Weeks	Excellent
Gonçalves [50]	National	28	24.7 ± 4.7 year	Competition	51 Matches	Excellent
Jones [51]	National	37	23.0 ± 4.0 year	Competition	79 Matches	Excellent
Martín-García [52]	National	24	20.0 ± 2.0 year	Training Competition	42 Weeks 37 Matches	Excellent
Owen [53]	National	29	26.7 ± 4.0 year	Training Competition	80 Sessions 20 Matches	Good
Owen [54]	National	20	26.7 ± 4.1 year	Training Competition	88 Sessions 22 Matches	Excellent
Rago [55]	National	14	27.6 ± 3.5 year	Competition	6 Matches	Excellent
Reche-Soto [56]	National	21	N/D ^1^	Competition	12 Matches	Excellent
Wiig [57]	National	75	20.4 ± 4.6 year	Competition	3 Matches	Good

^1^ Not defined (N/D).

**Table 3 ijerph-18-02721-t003:** Characteristics of studies that evaluated both internal and external load measures.

Study	Level	Sample	Age	Condition	Duration	Quality
Abade [58]	National	56 66 29	U14—14.0 ± 0.2 year U17—15.8 ± 0.4 year U19—17.8 ± 0.6 year	Training	12 Sessions 16 Sessions 10 Sessions	Excellent
Akenhead [59]	National	33	24.0 ± 4.0 year	Training	48 Sessions	Excellent
Aslan [60]	National	47	17.6 ± 0.58 year	Competition	4 Matches	Excellent
Azcárate [61]	National	20	27.1 ± 3.1 year	Training Competition	46 Sessions 10 Matches	Excellent
Brink [62]	National	16 15	U15—14.3 ± 0.3 year U17—16.3 ± 0.2 year	Training	40 Sessions 48 Sessions	Excellent
Casamichana [63]	National	28	22.9 ± 4.2 year	Training	44 Sessions	Excellent
Castagna [64]	National	1200	24.5 ± 0.8 year	Competition	60 Matches	Good
Condello [65]	Regional	17	24.9 ± 4.2 year	Training Competition	20 Sessions 4 Matches	Excellent
Coppalle [66]	National	26 24	26.2 ± 5.1 year 25.9 ± 5.2 year	Training Competition	12 Weeks ^1^	Excellent
Coutinho [67]	National	56 66 19	U15—14.0 ± 0.2 year U17—15.8 ± 0.4 year U19—17.8 ± 0.6 year	Training	12 Sessions 11 Sessions 10 Sessions	Excellent
Curtis [68]	National	18	20.0 ± 1.0 year	Competition	24 Matches	Excellent
Iacono [69]	National	24	18.3 ± 1.1 year	Training Competition	8 Sessions 14 Matches	Excellent
Figueiredo [70]	National	18	22.0 ± 2.0 year	Training	4 Sessions	Good
Fitzpatrick [71]	National	14	17.1 ± 0.5 year	Training Competition	23 Sessions 6 Matches	Excellent
Fullagar [72]	National	15	25.5 ± 4.9 year	Training Competition	5 Sessions 2 Matches	Excellent
Gaudino [73]	National	22	26.0 ± 6.0 year	Training	38 Weeks	Excellent
Geurkink [74]	National	46	25.6 ± 4.2 year	Training	61 Sessions	Excellent
Giménez [75]	National	14	23.2 ± 2.7 year	Competition	2 Matches	Excellent
Jaspers [76]	National	35	23.2 ± 3.7 year	Training Competition	2 Seasons	Excellent
Jaspers [77]	National	38	22.7 ± 3.4 year	Training	2 Seasons	Excellent
Malone [78]	National	30	25.0 ± 5.0 year	Training	45 Weeks	Excellent
Malone [79]	National	48	25.3 ± 3.1 year	Training	460 Sessions	Excellent
Malone [80]	National	30	25.3 ± 3.1 year	Training	240 Sessions	Excellent
Malone [81]	National	37	25.0 ± 3.0 year	Training Competition	48 Weeks	Excellent
Noor [82]	National	35	25.9 ± 3.8 year	Training Competition	16 Weeks	Excellent
Oliveira [83]	National	13	26.2 ± 4.1 year	Training Competition	20 Sessions 9 Matches	Excellent
Oliveira [84]	National	19	26.3 ± 4.3 year	Training	189 Sessions	Excellent
Op De Beéck [85]	National	26	23.2 ± 3.7 year	Training Competition	1 Season	Excellent
Owen [86]	National	10	26.8 ± 4.1 year	Training	8 Weeks	Excellent
Rago [87]	National	17	27.8 ± 3.9 year	Training Competition	67 Sessions 17 Matches	Excellent
Rago [88]	National	13	25.8 ± 3.5 year	Training Competition	42 Sessions 3 Matches	Excellent
Scott [89]	National	15	24.9 ± 5.4 year	Training	29 Sessions	Excellent
Silva [90]	National	20	26.5 ± 3.9 year	Training	15 Sessions	Excellent
Stevens [91]	National	28	21.9 ± 3.2 year	Training Competition	76 Sessions 3 Matches	Excellent
Suarez-Arrones [92]	National	30	N/D ^2^	Competition	2 Seasons	Excellent
Torreño [93]	National	26	27.3 ± 3.4 year	Competition	2 Seasons	Excellent
Zurutuza [94]	National	15	25.2 ± 3.0 year	Training Competition	20 Sessions 8 Matches	Excellent

^1^ Two seasons, 6 weeks in each season; ^2^ not defined (N/D).

## Data Availability

Data is contained within the article.

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
