# Peer review of "Load Measures in Training/Match Monitoring in Soccer: A Systematic Review"

_ijerph, 2021, doi:10.3390/ijerph18052721_

Round 1
Reviewer 1 Report
“The purpose of the present study is to carry out a systematic review on internal and external load monitoring, in training/match, identifying the measures used. Simultaneously, we intend to make recommendations that allow to standardize the classification and use of the different load measures.”
While I admire the effort put in this study, I did not feel that this study achieved its purpose. The most important question that need to be addressed in such a great review of the literature is how these different approaches serves in practical manner. In other words, why would the authors make the recommendations they made?
For instance: the authors recommended “the following measures should be considered: average heart rate (%HRMAX); heart rate intensity zones (min and 674 %min); sRPE (AU and AU/min); Edward’s Training Load (AU); Hooper Index (AU); Player Load 3D (AU, AU/min and %AU); Total Distance Covered (m, 678 m/min and %m); Distance covered by speed zone (m, m/min and %m); Distance ratios; Accelerations and decelerations(n, n/min and %n); and training/match duration (min).” how do the authors propose that their recommendations to be followed? Indicating “five internal load and six external load measures”?
Furthermore, I feel that the authors misinterpreted speed with acceleration claiming the same measuring unite should be used by converting the km/h to “m/s2”. How do the authors suggest that the speed km/h should be converted to “m/s2”. And the argument that m/s would be the same as “m/s2” is totally not correct.
Regarding the recommendations the authors made, they just converted the km/h to m/s? so? What’s new?
When classifying and arguing for the finding from other studies, like the once reported in Wrigley [36], Iacono [68], Malone [77], Silva [89] and Torreño [92], the authors ignored that the measured intensity in any match is depending on the tactical part of the match (i.e., defensive, offensive) the opponent playing style etc… therefore, irrespective of the classifications, there will always be difference.
We can also say that training load is a planned load whereas match load is unplanned load, depends (i.e., see my former comment).
All in all, I feel this study is a good update study that takes new research made after 2017 in account. However, the paper need to be systemized for better readability, so the reader can take the message. As a coach, I would be very amazed if I would be able to just capture one or two of the recommended measures.
Hence, I would like to invite the authors to look at the following “very good studies” and try to systemize their paper so it has a message.
- (PDF) Internal and External Training Load: 15 Years On (researchgate.net)
- (PDF) Monitoring Training Loads: The Past, the Present, and the Future (researchgate.net)
- Maybe helpful: Monitoring Training Load to Understand Fatigue in Athletes (nih.gov)
Finally, I would suggest that the authors work a little harder in sending a message to the readers.
Author Response
Dear Reviewer,
We sent a file with our responses to your comments. Once again, thanks for the collaboration that certainly helped us to improve the review article.
Best regards!
All the authors

Reviewer 2 Report
Was this study registered in PROSPERO or similar?
Why the authors used only web of science?
Explain more about the "association" between key words used in a research strategy in the database.
The explanation about the tool used to evaluate the quality of studies needs to be improved.
Please, to add the strength and the limitations of this systematic review.
Author Response

(The authors gave the same response as above.)

Round 2
Reviewer 1 Report
no further comments
Reviewer 2 Report
Congratulations.